# Assessment of a 4-Week Starch- and Sucrose-Reduced Diet and Its Effects on Gastrointestinal Symptoms and Inflammatory Parameters among Patients with Irritable Bowel Syndrome

**DOI:** 10.3390/nu13020416

**Published:** 2021-01-28

**Authors:** Clara Nilholm, Ewa Larsson, Emily Sonestedt, Bodil Roth, Bodil Ohlsson

**Affiliations:** 1Department of Internal Medicine, Skåne University Hospital, SE-20502 Malmö, Sweden; ewa.larsson@outlook.com (E.L.); bodil.roth@med.lu.se (B.R.); bodil.ohlsson@med.lu.se (B.O.); 2Department of Clinical Sciences, Lund University, SE-20502 Malmö, Sweden; emily.sonestedt@med.lu.se

**Keywords:** dietary intervention, IBS, inflammation, starch, sucrose

## Abstract

Dietary advice constitutes a treatment strategy for irritable bowel syndrome (IBS). We aimed to examine the effect of a starch- and sucrose-reduced diet (SSRD) on gastrointestinal symptoms in IBS patients, in relation to dietary intake and systemic inflammatory parameters. IBS patients (*n* = 105) were randomized to a 4-week SSRD intervention (*n* = 80) receiving written and verbal dietary advice focused on starch and sucrose reduction and increased intake of protein, fat and dairy, or control group (*n* = 25; habitual diet). At baseline and 4 weeks, blood was sampled, and participants filled out IBS-SSS, VAS-IBS, and Rome IV questionnaires and dietary registrations. C-reactive protein and cytokines TNF-α, IFN-γ, IL-6, IL-8, IL-10, and IL-18 were analyzed from plasma. At 4 weeks, the intervention group displayed lower total IBS-SSS, ‘abdominal pain’, ‘bloating/flatulence’ and ‘intestinal symptoms´ influence on daily life’ scores (*p* ≤ 0.001 for all) compared to controls, and a 74%, responder rate (RR = ΔTotal IBS-SSS ≥ −50; RR_controls_ = 24%). Median values of sucrose (5.4 vs. 20 g), disaccharides (16 vs. 28 g), starch (22 vs. 82 g) and carbohydrates (88 vs. 182 g) were lower for the intervention group compared to controls (*p* ≤ 0.002 for all), and energy percentages (E%) of protein (21 vs. 17 E%, *p* = 0.006) and fat (47 vs. 38 E%, *p* = 0.002) were higher. Sugar-, starch- and carbohydrate-reductions correlated weakly-moderately with total IBS-SSS decrease for all participants. Inflammatory parameters were unaffected. IBS patients display high compliance to the SSRD, with improved gastrointestinal symptoms but unaltered inflammatory parameters. In conclusion, the SSRD constitutes a promising dietary treatment for IBS, but needs to be further researched and compared to established dietary treatments before it could be used in a clinical setting.

## 1. Introduction

Irritable bowel syndrome (IBS) is a functional gastrointestinal (GI) disorder with varying prevalence estimates of 4–11% depending on geography and the diagnostic criteria used [1,2]. It is characterized by abdominal pain and altered bowel habits, in the absence of identifiable organic disease. Of IBS patients, 62–90% attribute aggravation of GI symptoms to intake of specific foods [3]. Accordingly, diet modification is employed as a primary treatment strategy. The National Institute for Health and Care Excellence (NICE) guidelines, which recommend regular meal patterns and decreased intake of mineral water, caffeine, fat and spicy foods, are used as a first-line treatment for IBS [4]. As a second-line treatment, the low FODMAP diet, which advocates exclusion of fermentable oligo-, di- and monosaccharides and polyols, is an evidence-based dietary treatment commonly used in the clinical setting [5]. Still, 20–50% of IBS patients do not experience reduced GI symptom burden following NICE guidelines and/or a low FODMAP diet [6]. As such, research is needed to identify symptom-inducing foods in IBS, in order to develop alternative, individualized dietary treatments.

After initial digestion by mastication and salivary amylase, starch and sucrose are primarily digested by the sucrase-isomaltase (SI) enzyme in the ileum [7]. Previous research has identified an increased prevalence of rare SI gene variants in IBS patients [8,9], which confer reduced enzymatic activity and subsequent insufficient starch and sucrose digestion [10]. Unabsorbed carbohydrates may induce digestive symptoms through colonic fermentation or osmotic activity, resulting in bloating, flatulence, abdominal pain, and diarrhea [3,5]. Historically, clinical investigation of IBS patients has focused on identifying fructose and/or lactose intolerance, conditions identified in up to 70% of IBS patients [11]. In a recent study by Kim et al. [12], SI deficiency (SID) was identified through intestinal biopsy in 35% of patients with presumed IBS-D/M. These data warrant further study of the relevance of starch and sucrose intolerance in IBS.

High dietary sugar intake has been proposed to increase subclinical inflammation, as shown by elevated C-reactive protein (CRP) in some previous studies [13]. Additionally, intake of added sugar and sugar-sweetened beverages is associated with increased expression of proteins involved in inflammatory response signaling [14]. Diets rich in sugar, starch and saturated fats could activate the innate immune system through increased production of pro-inflammatory cytokines [15]. An association between IBS and elevated amounts of circulating cytokines, such as IL-6, IL-8, TNF-α, and IL 1β has been documented, although these results were not reproducible in other studies [3].

We have previously reported that IBS patients markedly improve their GI symptoms following a starch- and sucrose-reduced diet (SSRD) for 2 weeks [16]. Two hypotheses were raised: (1) the SSRD has continued positive effects on GI symptoms in IBS patients after 4 weeks and (2) improvements are related to a quantifiable decrease in starch and sucrose intake and reduced plasma CRP and cytokine levels. The primary aim of the present study was to evaluate the effect of the SSRD on IBS patients after 4 weeks. Secondary aims were to quantify and correlate significant changes in nutrient intakes and systemic inflammatory CRP and cytokine levels with GI symptom changes.

## 2. Materials and Methods

### 2.1. Study Design and Subjects

The study was a randomized, open clinical trial with a 4-week dietary intervention in IBS patients. Patients with an IBS diagnosis, 18–70 years, with Northern European ancestry, were recruited from registries from Primary Healthcare Centers (PCC) (2015–2017) and the Department of Gastroenterology (2016–2017), Skåne University Hospital, according to the International Statistical Classification of Diseases and Related Health Problems–ICD-10.

The recruitment process (Figure 1 and Appendix A) is described in a previous publication [16]. Initially, 2034 IBS patients from PCC and 789 IBS patients from the Department of Gastroenterology were identified. After exclusion of 1144 duplicates, 1679 patients remained. Invitation letters were sent to 528 randomly selected patients from the PCC and 151 patients from the tertiary healthcare center after exclusion of patients who had moved from the area or who had no available phone number. In a random order, patients were contacted via regular mail with information about the study. Afterwards, follow-up calls were made to contacted patients and 145 were willing to participate (33 patients [23%] from the tertiary healthcare center; 111 women [77%]).

Questionnaires covering sociodemographic and lifestyle factors, the Rome IV questionnaire, the irritable bowel syndrome-severity scoring system (IBS-SSS), the visual analog scale for irritable bowel syndrome (VAS-IBS), the Bristol stool form scale (BSFS), and food diaries were sent to the 145 patients for completion during the run-in period, before an appointment at the Department of Internal Medicine. At the appointment, all patients were thoroughly examined by a physician and a medical history was taken. Anthropometric measurements of weight (kg) and height (m) in light in-door clothing, and systolic and diastolic blood pressures (mmHg) (Omron^®^ automatic reading, Omron Electronics AB, Malmö, Sweden) in supine position were performed at study start, as well as after the study completion. The inclusion period lasted from January 2018 to February 2019.

Inclusion criteria for the study were a diagnosis of IBS, age 18–70 years, and Northern European heritage, i.e., with Scandinavian or Northern European-born parents and grandparents. Exclusion criteria were insufficient symptoms, <175 score on total IBS-SSS, presence of any organic GI disease, severe organic and psychiatric diseases, or already on a diet, i.e., vegan diet, gluten-free diet, low carbohydrate high fat (LCHF) diet, or low FODMAP diet (Appendix A).

Forty patients were excluded because they withdrew their consent to participate, had a wrong diagnosis, were already on a diet, or had insufficient symptoms (Figure 1). From a total of 679 invitation letters sent out, 105 patients (77 patients [73%] recruited from primary care and 28 patients [27%] from the tertiary health center); 82 women [78%] and 23 men [22%]; were finally included in the study, with a 15% inclusion rate.

Eighty patients were randomly selected to the SSRD, whereas 25 participants served as controls, following their habitual diet. The BSFS was completed daily during the study period. After 4 weeks, the IBS-SS and VAS-IBS questionnaires were repeated. Blood samples were collected before and after the intervention, and the plasma was separated and kept at −80°C until analysis. Ninety-seven (97) patients completed the study.

Blood samples from 105 age-matched non-IBS controls (median age: 48 [range, 20–67] years, 45% women, mean body mass index (BMI): 23.1 ± 4.3 kg/m^2^) with no prior diagnosis of organic or functional disease or present GI complaints, were collected from the Malmö Offspring study (MOS) registry [17] and served as controls for baseline cytokine analyses. The MOS is a population-based study including subjects from the general population after an anthropometric examination and completion of questionnaires [18]. The study started in 2013 and has so far included around 4500 participants. The current controls were enrolled during 2015.

### 2.2. Dietary Advice

Patients received verbal as well as written dietary advice focused on starch- and sucrose reduction, and increased intake of certain fruits and vegetables, meat, fish, and dairy products. The dietary advice was modified from dietary guidelines for patients with congenital sucrase-isomaltase deficiency (CSID) [19]. Briefly, all sucrose-containing foods were to be avoided. Processed rice and pasta were discouraged and fiber-rich alternatives (≤1 serving daily) were allowed. Fiber-rich bread was recommended instead of white bread. Whole grains from oats, barley and bran were recommended instead of processed breakfast cereals. Pork, beef, lamb, fish, turkey, chicken, and egg could be ingested without any restrictions. Processed meat such as bacon, sausage and pies should be avoided if they were treated with sugar or starch. Dairy products could be ingested without restrictions, but natural products without added sugar were recommended. Butter and oil intake was unrestricted, but margarine should be avoided. Drinks in the form of milk, sugar-free soda and home-made juices were allowed, if the juice was made from recommended fruits. Regarding spices, salt, pepper and fresh herbs could be used unrestrictedly. The participants were encouraged to read the table of contents carefully for all products. Nuts and seeds were recommended in place of sugary snacks. Increased fat and/or protein intake and prolonged chewing was encouraged, to enhance salivary amylase breakdown of starch and delay gastrointestinal transport. No advice was provided regarding food intake frequency or regularity. Patients were provided with visual aids to familiarize themselves with allowed and prohibited foods, including lists of suitable fruits and vegetables with less starch and sucrose content (Appendix A, Appendix A). Participants in the control group received no dietary advice and were urged not to make any changes to their ordinary diet.

All participants were encouraged to continue with their ordinary energy intake, degree of physical activity and medications, without making any changes. If they used any form of probiotics or were on a diet not excluding them from enrollment, they should continue with this during the study, without introducing any new drugs or other dietary changes. The participants could reach the study staff by telephone or email whenever they wanted during the study.

### 2.3. Questionnaires

#### 2.3.1. Study Questionnaire

A study questionnaire about sociodemographic factors, family history, lifestyle habits, medical health, and pharmacological treatment was completed prior to study start. This questionnaire was similar in both the current interventional study and in the MOS [18]. In the MOS questionnaire, the participants were asked: “Have you experienced GI symptoms during the past 2 weeks?”. If they answered “yes” to this question, they were encouraged to complete the VAS-IBS (see below). If they answered “no”, they did not complete the VAS-IBS.

#### 2.3.2. Food Diary Registrations

Participants were instructed to keep a dietary record of all consumed foods, in a free writing structure, with the requirement to provide information on eating habits, time/type of food intake and GI symptoms in relation to food intake for 4 days, before and at the end of the dietary intervention. The patients reported the amount and/or volume of each food item, including the percentage of fat in dairy products, fiber in bread products and cacao in chocolate, as well as information on the type of soda (sugar-free or regular) consumed. The manufacturer of the product was given when applicable, e.g., for brands of bread, butter, and muesli. The product name and ingredient list for pre-maid dishes was reported. For each patient, nutrient intake was calculated from a single day (day 2) of the 4-day registrations. Daily nutrient intake calculations, in total amounts of grams and energy percentages (E%), were performed by a nutritionist, using the AIVO Diet computer program from the National Food Agency, Sweden [20]. The percentages of participants who had an intake of micronutrients equal to or above the average intake requirement (AR = the nutrient intake level meeting the requirements of 50% of the population) were calculated based on dietary reference intakes values from the 2012 Nordic nutrition recommendations when available [21]. Nutrient intake of non-IBS controls was gathered from the MOS data base. The intakes were calculated from a web-based 4-day food record designed by the Swedish National Food Agency, where controls registered everything they ate and drank [18].

#### 2.3.3. Rome IV Questionnaire

The Swedish version of the Rome IV questionnaire (Questions No. 40–48) was used to diagnose functional gastrointestinal disease (FGID) [22]. License for usage was obtained from the Rome Foundation, Inc., Raleigh, NC, USA.

#### 2.3.4. The Irritable Bowel Syndrome-Symptom Severity Score Questionnaire

The IBS-SSS is a validated questionnaire for assessing IBS severity [23]. Using visual analog scales, it covers abdominal pain, abdominal distension, bowel habit satisfaction and the impact of bowel habits on daily life. The score ranges between 0 mm (‘no symptoms’), and 100 mm (‘severe symptoms’). Abdominal pain frequency in the last 10 days is registered. The combined maximum total score is 500. Scores of 75–174 indicate mild, 175–299 moderate, and ≥300 severe disease [23].

#### 2.3.5. The Visual Analog Scale for Irritable Bowel Syndrome Questionnaire

The VAS-IBS is a validated symptom-assessment questionnaire [24] covering abdominal pain, diarrhea, constipation, bloating and flatulence, vomiting and nausea, psychological well-being, and intestinal symptoms’ influence on daily life. Items are measured on a scale of 0–100 mm, ranging from absent (0 mm) to very severe (100 mm) symptoms. The values are inverted from the original format, where 100 mm indicated ‘no symptoms’. Reference values of VAS-IBS in a healthy population were previously determined by a control material consisting of 90 healthy volunteers (median age: 44 [range, 21–77] years, 57% women) included from staff at the Skåne University Hospital, Department of Internal Medicine, during 2010 [25].

Participants in the MOS could not be used as controls for VAS-IBS variables as they were instructed not to complete the VAS-IBS questionnaire if they negated presence of GI symptoms during the past 2 weeks.

#### 2.3.6. Bristol Stool Form Scale

Participants registered bowel habits continuously during the 10-day run-in period and 4-week dietary intervention. Stool frequency and consistency was reported according to the previously validated BSFS [26].

### 2.4. CRP and Cytokine Analyses

The Mesoscale Discovery^®^ (MSD) V-plex Plus Proinflammatory Panel 1 Human Kit (Rockville, MD, USA) was used to perform the cytokine analyses by electro-chemiluminescence detection (Lot No: K0081301) [27]. A multiplex assay was used for cytokines TNF-α, IFN-γ, IL-1β, IL-2, IL-4, IL-6, IL-8, IL-10, IL-12p70, and IL-13. The inter-assay coefficient of variation (CV) was below 15% and the intra-assay CV below 7%. IL-18 was analyzed by a separate U-plex assay (Lot No: 319780) [28]. All samples from both IBS and non-IBS patients were analyzed simultaneously at the laboratory using the same lot numbers.

MSD GOLD Small Spot Streptavidin plates pre-coated with the respective diluted biotinylated capture antibody were washed three times with MSD wash buffer. In each well, 50 µL of serially diluted calibrator samples and patient plasma (thawed from storage in −80 °C) were added and the plates sealed. The plates were then incubated at room temperature (RT) for 2 h with shaking. The washing procedure was repeated and 50 µL of the respective SULFO-TAG conjugated detection antibody solution was added to the wells. After another incubation and washing procedure, 150 µL of MSD read buffer T was added to each well and the plates read immediately using an MSD instrument.

The cytokines IL-1β, IL-2, IL-4, IL-12p70, and IL-13 were excluded from data analysis due to high CVs and/or many samples with values below detection levels.

Plasma levels of CRP were analyzed according to clinical routines at the Department of Clinical Chemistry. Reference values of the laboratory were used for classification of abnormal values [29].

### 2.5. Statistical Analyses

Calculations were performed in SPSS (version 26; IBM Corporation, Armonk, NY, USA) as the intention to treat. *p* < 0.05 was considered statistically significant.

#### 2.5.1. Study Group Size

Since this was a pilot-study investigating the effects of the SSRD on IBS patients, no power analysis was performed due to lack of previous data. However, our research team has previously investigated the effect of a carbohydrate-reduced diet on type 2-diabetes patients, where 23 participants demonstrated improved GI symptoms and changes in systemic cytokine expression after 12 weeks [30,31]. As the dietary intervention in the present study was of a shorter duration, we hypothesized that more participants would be required to show a possible effect of the SSRD.

Additionally, although not discussed in the present paper, the complete study scope included genetic testing for a rare SI functional gene variant [8,9], which influenced the intervention/control group ratio as we also planned to stratify the intervention group according to gene variant.

#### 2.5.2. Non-Parametric Statistical Analyses

Tests for normality were performed by the Kolmogorov–Smirnov test.

The majority of the cytokine, nutrient and GI symptom score variables were non-normally distributed at both time-points, therefore, results are presented as median values with interquartile ranges. Mann–Whitney U test and Fisher’s exact test were used for between-group comparisons of continuous and categorical variables, respectively. Wilcoxon test was used for within-group-comparisons of changes from baseline. Baseline variables for IBS patients and non-IBS controls were adjusted for BMI and sex using Generalized Linear Model. Subgroup analyses were likewise performed using Generalized Linear Model. Results are presented as β-values and 95% confidence intervals (CI). Correlations were performed between changes in total IBS-SSS/VAS-IBS scores and changes in protein, fat, carbohydrate, fiber, starch, sucrose, total sugar, disaccharide, and monosaccharide intakes. GI symptom and nutrient delta values (baseline to 4 weeks) were both normally and non-normally distributed. Spearman’s correlation test was used for non-normally distributed variables.

#### 2.5.3. Parametric Statistical Analyses

Pearson’s correlation test was employed for normally distributed GI symptom and nutrient delta values. ANOVA and post-hoc Bonferroni were used for IBS subgroup analyses of GI symptom score delta values. Results are presented as means ± standard deviations.

#### 2.5.4. Effect Size Calculations

The clinical effect size for between-group analysis was evaluated with non-parametric methods. Eta squared (η^2^) was calculated from the Z distribution produced in the Mann-Whitney U test, by the formula r^2^ = Z^2^/*n*. The index assumes values from 0 to 1 and multiplied by 100% indicates the percentage of variance in the dependent variable explained by the independent variable [32]. Cliff’s delta value was calculated in Excel. This score ranges from −1 to 1; where 1 indicates that all observations from the intervention group are greater than all observations from the control group, and −1 indicates the opposite situation. The value of zero (0) indicates equality of observations between groups [33].

## 3. Results

### 3.1. Participant Characteristics

For all IBS patients included in the dietary intervention study (*n* = 105), the median age was 46 (34.5–57) years and median BMI was 24.5 (22.4–27.7) kg/m^2^ (range 16.0–39.8 kg/m^2^). Median disease duration was 18.5 (10.0–29.0) years (range 3.0–60.0 years).

The age and weight of participants differed significantly between the intervention and control group at baseline, with lower median values in the control group (Table 1). There were no significant differences between groups for other characteristics, i.e., disease duration (Table 1) or sex, BMI and level of physical activity (*p* = 0.14, *p* = 0.23 and *p* = 0.44; Appendix A).

The most frequent comorbidities among IBS patients were allergies, hypothyroid disease, asthma, depression and hypertension, and the most commonly used medications were antidepressants, levothyroxine, laxatives, and proton pump inhibitors (Appendix A).

### 3.2. Gastrointestinal Symptoms and IBS and Non-IBS FGID Subgroups at Baseline

According to Rome IV criteria, 37 participants (35.2%) displayed mixed IBS (IBS-M), 26 participants (24.8%) diarrhea-predominant IBS (IBS-D), 20 participants (19.0%) constipation-predominant IBS (IBS-C), and 3 participants unspecified IBS (IBS-U). In 17 patients (16.2%), abdominal pain and bowel habits were not closely associated, rendering the diagnosis of non-IBS FGID. Two patients in the intervention group did not complete the Rome IV questionnaire. The distribution of IBS subgroups did not differ significantly between the intervention and control group (Appendix A).

Forty-eight subjects (47.6%) had moderate and 55 subjects (52.4%) severe disease symptoms according to total IBS-SSS scores.

### 3.3. Gastrointestinal Symptoms after the 4-Week Dietary Intervention

Baseline scores of total IBS-SSS and VAS-IBS symptoms did not differ significantly between the intervention and control group (Table 1). At 4 weeks, a decrease in total IBS-SSS, from a median value of 306 to 156, was observed in the intervention group (corresponding change for controls: 310 to 300 points) (Table 1). Seventy-four percent (74%) of patients in the intervention group and 24% in the control group were classified as responders to the diet, as defined by a decrease in total IBS-SSS of ≥50 points [23]. Results showed medium-to-large effect sizes in favor of the dietary intervention group (η^2^ = 0.20, Cliff’s δ = 0.70).

Some VAS-IBS symptom scores differed between the intervention group and control group at 4 weeks, i.e., ‘Abdominal pain’ (*p* < 0.001), ‘Bloating and flatulence’ (*p* < 0.001), ‘Vomiting and nausea’ (*p* = 0.04) and ‘Intestinal symptoms’ influence on daily life’ (*p* < 0.001). VAS-IBS scores of ‘Diarrhea’, ‘Constipation’ and ‘Psychological well-being’ did not differ significantly between groups (Table 1). Average daily stool frequency did not differ significantly between groups at baseline or 4 weeks (Table 1).

The distribution of changes (median differences) in total IBS-SSS score and VAS-IBS scores of ‘Abdominal pain’, ‘Diarrhea’, ‘Bloating and flatulence’ and ‘Intestinal symptoms’ influence on daily life’ differed significantly, with greater median score decreases in the intervention group (Table 1 and Figure 2a,b). Remaining VAS-IBS scores did not change in a significantly different manner between groups (Table 1).

### 3.4. Dietary Intake

Dietary intakes did not differ significantly between the intervention and control group at baseline with regards to energy content, intake of major food groups (Table 2), sugar and starch intake (Table 3), and micronutrient intake (Appendix A). A high percentage of patients had vitamin intake levels below the AR at baseline, most prominent for vitamin D and riboflavin. For thiamin, niacin, vitamin B6, folic acid and vitamin C, a range between 52–75% of patients met the AR in both groups (Appendix A). For minerals phosphate, calcium and zinc, 68–100% of patients reached the AR in both groups at baseline, whereas iron, iodine and selenium intakes were lower (Appendix A). The intervention produced no significant changes in micronutrient intakes for either the intervention or control group (Appendix A).

At 4 weeks, carbohydrate intake in gram and E% differed significantly between the intervention and control group (*p* < 0.001 for both) (Table 2). Protein and fat intakes (E%) were significantly higher in the intervention group compared to controls (Table 2). Total energy content did not differ significantly between groups (Table 2).

At 4 weeks, sucrose, disaccharide, and starch intakes were all significantly lower within the intervention group, compared to controls (Table 3).

Sucrose, disaccharide, total sugar and starch intakes (grams and E%) all decreased significantly in the intervention group (*p* < 0.001 for all), but not in controls (*p* ≥ 0.14 for all). Out of the sugars, only monosaccharide intake did not decrease in the intervention group during the study period (*p* = 0.13 for g and *p* = 0.78 for E%).

#### 3.4.1. Evaluation of Compliance

In the intervention group, 80% of participants (*n* = 64) lowered their starch intake from baseline, as compared to 48% (*n* = 12) in the control group. The intervention group lowered their starch intake more than the control group, with 64% of participants decreasing their starch intake with >50%, compared to 24% for controls. Lowered sucrose intake levels at 4 weeks were similarly more frequent in the intervention versus control group (78%, *n* = 62 vs. 60%, *n* = 15). Additionally, 68% (*n* = 54) of participants in the intervention group and 28% (*n* = 7) in the control group lowered their sucrose intake with >50% from baseline.

#### 3.4.2. Dietary Intake in Non-IBS Controls

The dietary intake of non-IBS controls differed from that of IBS patients at baseline. Energy intake was significantly higher in controls, as were energy percentages of protein, starch and disaccharides. Carbohydrate, total fat and monosaccharide energy percentages were similar between groups (Appendix A).

### 3.5. Correlations between Changes in Gastrointestinal Symptoms and Dietary Intakes

Weak-moderate positive correlations between changes in total IBS-SSS and carbohydrate and starch intake (Figure 3 and Appendix A) as well as sucrose, disaccharide and total sugar intake reductions (Appendix A) were observed.

Out of all the VAS-IBS scores, significant correlations were identified for improvement in ‘Bloating and flatulence’, which correlated positively with reductions in carbohydrates and starch, and improvement in ‘Intestinal symptom’s influence on daily life’, which correlated positively with reductions in carbohydrate and disaccharide intakes (Appendix A). No significant correlations were identified for other VAS-IBS scores. Changes in GI symptoms did not correlate with changes in protein, fat, fiber or monosaccharide intake (Appendix A).

### 3.6. Plasma Concentrations of CRP and Cytokines

CRP levels in plasma did not differ between groups and were not affected by the dietary intervention (Table 4). TNF-α levels differed between IBS patients and non-IBS controls (2.47 [2.07–2.89] pg/mL vs. 0.82 [0.63–1.05] pg/mL; β: 4.3; 95% CI: 1.5–7.1; *p* = 0.003) at baseline. TNF-α levels in the IBS group did not correlate significantly with any GI symptoms at baseline (*p* ≥ 0.084 for all correlations). For the remainder of the cytokines (IFN-γ, IL-6, IL-8 and IL-18 and the anti-inflammatory IL-10), no significant differences between IBS and non-IBS subjects were found (*p *≥ 0.15 for all).

Cytokine levels did not differ significantly between the intervention and control group at baseline or 4 weeks, but approached significance for IL-8 at 4 weeks (*p* = 0.054 for IL-8 and *p* ≥ 0.078 for remaining cytokines). IL-8 levels increased within the intervention group, and TNF-α levels increased in the control group (Table 4).

The changes in IL-8 levels in the intervention group, and in TNF-α levels in the control group, did not correlate with any changes in GI symptom scores or nutrient levels in the respective groups (*p* ≥ 0.071 and *p* ≥ 0.28, respectively).

### 3.7. Subgroup Analysis of GI Symptoms

Regression analyses were performed with IBS-C, IBS-D, IBS-M, and non-IBS FGID subgroups with respect to GI symptom changes, excluding the small IBS-U subgroup (*n* = 3). In the intervention group, results showed a near-significant difference between group means for change in total IBS-SSS score (*p* = 0.052), and significant differences for ‘Diarrhea’ (*p* < 0.001), ‘Constipation’ (*p* < 0.001), ‘Bloating and flatulence’ (*p* = 0.03) and ‘Intestinal symptom’s influence on daily life’ (*p* = 0.038). Post-hoc analysis revealed a near-significant greater improvement in total IBS-SSS for IBS-M compared to IBS-C patients (*p* = 0.054). The IBS-C group differed from the IBS-D and IBS-M groups in terms of improvement of constipation and diarrhea (Table 5).

Improvement in ‘bloating and flatulence’ was significantly greater for the IBS-M group compared to the IBS-D and non-IBS FGID groups (Table 5). The ‘Intestinal symptoms’ influence on daily life’ score improved significantly more for the IBS-M group compared to the IBS-C group (Table 5). No significant differences were found between subgroups in the control group (*p* ≥ 0.20 for all GI symptoms).

## 4. Discussion

This is the first dietary intervention study evaluating the direct effect of starch and sucrose reduction in IBS. The main finding was that IBS patients showed high compliance to the SSRD for 4 weeks, with markedly reduced GI symptoms. Reductions in starch, sucrose, disaccharide and carbohydrate intakes correlated with improvement of total IBS-SSS and/or specific GI symptoms. However, the SSRD has a minor effect on circulating CRP and cytokine levels.

The observation of low micronutrient intakes suggests a nutrient-lacking diet among study participants, which is in line with the infrequent fruit- and vegetable intakes and insufficient iron- and vitamin D blood levels previously reported for this group [16]. Taken together, this marks the importance of reviewing basic dietary habits in IBS patients, as has been previously emphasized [5].

Starch and sucrose belong to the overarching family of carbohydrates, which together with proteins and fats make up the three major macronutrients in the human diet [7,12]. Limited previous data suggests an association between a high intake of carbohydrates and aggravation of GI symptoms in IBS patients. A large French cohort study surveying 2423 IBS patients identified a moderate risk increase of IBS for patients following a sugar- and starch-rich ‘western dietary pattern’, although this diet was also high in fats [34]. Austin et al. [35] reported decreased abdominal pain and diarrhea and improved quality of life in IBS-D patients after 2 weeks on a very low-carbohydrate diet (VLCD) with 20 g carbohydrates per day (4 E%). The low FODMAP diet, which excludes or minimizes intake of lactose, sorbitol and fructose, without restriction of either starch or sucrose, has, however, been the research focus of dietary management of IBS for more than the last decade [3]. Thus, the role of starch and sucrose in IBS symptomatology and pathogenesis has yet to be clearly elucidated.

The digestion of starch begins by mastication and the digestive action of salivary α-amylase, resulting in smaller glucose polymers, which are further processed in the duodenum by pancreatic α-amylase and a-glucosidases. The resulting breakdown products dextrin and maltose, as well as sucrose, are hydrolyzed by the SI enzyme into monosaccharides glucose and fructose, in the brush-border of the enterocytes. Glucose and fructose are then absorbed by way of transmembrane-protein glucose transporters GLUT-2 and/or sodium-glucose cotransporter 1 (SGLT-1) [7,36]. Carbohydrate intolerance has been previously implicated in FGID. Mainly, fructose and lactose intolerance syndromes can be diagnosed via the hydrogen breath test, which measures hydrogen (or methane) in the outbreath after provocation with fructose/lactose. Sucrose intolerance can similarly be diagnosed with sucrose provocation [7]. As mentioned earlier, previous research has identified an increased prevalence of rare SI gene variants in IBS patients [8,9], which confer reduced enzymatic activity and subsequent insufficient starch and sucrose digestion [10]. The present results, showing correlations between decreased intakes of starch, sucrose, disaccharides and carbohydrates and improved GI symptoms, support a possible role of starch and/or sucrose intolerance. Unabsorbed starch and sucrose could increase small intestinal water volume, causing GI symptoms further exacerbated by visceral hypersensitivity in IBS patients, similarly to the proposed effects of FODMAPs [37]. Considering the high response rate of 74% observed in this study, other mechanisms besides those stemming from rare genetic abnormalities must be considered. Primarily, excess intake of starch and/or sucrose could be suspected to be responsible for the exhaustion of normal physiological systems responsible for their degradation. Still, starch and sucrose intolerance, induced either by genetic abnormalities or occurring due to other etiologies, should be considered possible mechanisms behind disease, at least in a subset of IBS patients.

As briefly mentioned in the introduction, a previous pilot-study showed an incidence of SID, diagnosed by the golden standard method of duodenal biopsy, in 11 (35%) out of 31 presumed IBS-D/M patients [12]. Interestingly, none of the 11 patients showed abnormal or heterozygous genetic testing and all displayed a concomitant lactase deficiency [12]. In the present study, where dairy intake was encouraged, the decreased intake of starch and sucrose alone still had a moderate-large positive effect on total IBS-SSS. The lower intake of starch and disaccharides observed in IBS patients compared to non-IBS controls at baseline, may be explained by a tendency of IBS patients to adjust their dietary habits and decrease their carbohydrate intake, as a form of self-medication.

Intestinal damage or dysbiosis could contribute to malabsorption and, although organic diseases such as celiac disease and inflammatory bowel disease were ruled out, participants were not investigated for other conditions, such as small-intestinal bacterial overgrowth (SIBO) [38]. A large meta-analysis of 25 case-control studies concluded prevalence of the condition in as many as 31% of IBS patients [38]. Dietary manipulation could affect the microbiota and improve SIBO [38,39]. A previous report demonstrated that dietary interventions for specific, malabsorbed carbohydrates influenced the composition of the gut microbiota [11].

Of individual GI symptoms, ‘Abdominal pain’, ‘Diarrhea’ and ‘Bloating and flatulence’ differed the most in degree of improvement between the intervention group and controls. Of note, these constitute the most common symptoms in patients with CSID [10]. Subgroup analysis revealed that the IBS-M subgroup tended to have greater effect on total-IBS SSS and Diarrhea’, ‘Constipation’, ‘Bloating and flatulence’, and ‘Intestinal symptoms’ influence on daily life’ as compared to other subgroups. In the future, the specific clinical features of IBS and non-IBS FGID patients who respond to the SSRD should be better outlined by examining larger study groups, in relation to the existence of possible SID.

Gluten and other non-gluten proteins in wheat, such as amylase-trypsin inhibitors or wheat-germ agglutinin, or the carbohydrate fructan, could also induce symptoms [40,41]. The amounts of these substances were presumably decreased following starch reduction, which could partly explain the improvements. Endoscopic wheat administration compromises the intestinal mucosa, leading to increased gut permeability [42]. Undigested wheat proteins could then be absorbed into the submucosa and activate resident innate immune cells. It has also been hypothesized that a diet with high sugar intake promotes inflammation [15]. However, most intervention trials have compared the effects of fructose versus glucose, and not sucrose, intake. A meta-analysis of these studies could not identify any differences in CRP levels between different sugar intakes [13]. In an epidemiological study, no associations between sugar or sugar-sweetened beverages and CRP could be observed. The associations between some proteins involved in inflammatory response and sugar could not confirm causality, but could rather be a marker of high sugar intake and risk of type 2 diabetes [14].

In the present study, where intakes of starch, sucrose and presumably wheat were reduced, systemic inflammatory cytokine concentrations and CRP remained unchanged Thus, the beneficial effect of the SSRD on GI symptoms does not appear to be mediated by decreased systemic inflammation. This, however, does not exclude a possible alteration of local submucosal inflammation, the determination of which would have required small intestinal biopsy. An alternative explanation for the absent change in cytokine levels lies in the patient population investigated. A recent population-based study showed that IBS subgroups identified by specific symptom-profiles differ in degree of healthcare utilization [43]. The current cohort, recruited mostly from primary care, could pertain to a different group than IBS patients from specialist gastroenterology units [43]. Speculatively, patients seeking specialist care could display more severe symptoms from specific underlying mechanisms, e.g., immune system activation. Previous studies have been inconclusive in determining whether circulating cytokine levels are actually elevated in IBS [3,44]. Review studies, however, conclusively identify elevated plasma TNF-α levels in IBS compared to non-IBS individuals [44,45], which is in line with our results.

### Study Strengths and Limitations

A strength of the SSRD is its relative simplicity compared to the low FODMAP diet. Although low FODMAP is effective for many IBS patients, it is laborious to implement, requiring a long follow-up period with re-introduction of tolerable FODMAPs. Further, studies usually evaluate the effect during complete FODMAP elimination, and not during the reintroduction phase [5]. The SSRD resembles a low-carbohydrate high fat (LCHF) diet, a term with which patients are often already familiarized. From our experience, the majority of included patients quickly grasped which foods to avoid when receiving the dietary advice. Solely decreasing intake of sucrose to low levels and carbohydrates to moderate amounts results in greatly decreased starch and sucrose intake overall. Such a dietary adjustment should be possible to maintain over time, whilst minimizing risk of malnutrition.

The present study has some limitations. Primarily, although a dietician assisted in study planning and nutrient intake calculations, there was no on-going dietician support. Study participants could contact the study staff, who could relay questions to the dietician. A few participants had trouble adjusting their diet and suffered minor weight loss. The set-up of the study, however, better simulates real-life conditions, and the high compliance observed suggests that the diet is easy-to-follow. An additional limitation is that nutritional intake was calculated from a single day at baseline and after 4 weeks. Micronutrient intakes in particular are difficult to estimate, as daily intakes can vary to a high degree, albeit in the studied population, the day-to-day variation of consumed foods was quite low. The complete change in participants’ diet means that specific foods causing the effects cannot be determined. However, the significant starch and sucrose reductions in the intervention group, and the correlations between improved GI symptom scores and starch and sucrose reductions for all participants, support our original hypothesis.

There may be a placebo effect, as participants expected improved GI symptoms when changing their diet. It is not possible to completely account for the inherent difference in expectations of benefit between groups. The use of a control group, with the same access to healthcare and contact availability with the study staff as the intervention group, partly adjusted for this effect. Additionally, there is a risk of self-selection bias, as patients participated voluntarily. The intervention to control group ratio in the current study was disproportionate with a smaller control group, and age and weight distribution differed between the two groups. The within-group statistical calculations, showing decreased intakes of starch and sucrose only in the intervention group but not in controls, were performed to partly compensate for any possible biases.

The non-IBS controls had higher carbohydrate and energy intake than the IBS patients, which could have influenced cytokine levels. However, slight differences in the methods of dietary record between studies may contribute to the observed differences as well. Additionally, IBS patients who are going to be included in a dietary trial might underreport or reduce their intake of prohibited foods just before entering the study. Furthermore, we were unable to compare levels of physical activity between groups due to usage of differing categorical variables in the questionnaires. The groups were, however, age-matched and calculations were adjusted for sex and BMI, factors which might be of greater importance for inflammation [15].

## 5. Conclusions

In conclusion, IBS patients show high compliance to a starch- and sucrose-reduced diet, with marked improvement of GI symptoms. The positive effect on GI symptoms is not mediated by reduced systemic inflammation. Alternative mechanisms at play, such as exhaustion of normal physiological systems, intestinal dysbiosis or SID could be areas for future research. The SSRD shows promise as a dietary treatment for IBS patients. Future studies should focus on (1) validating the SSRD as a dietary treatment for IBS through comparison with established dietary treatments and guidelines, i.e., the low FODMAP diet and NICE guidelines, (2) determining its efficacy and safety by examining long-term effects on GI symptoms and nutritional status, and (3) identifying pathogenetic mechanisms behind the improvements observed.

## Figures and Tables

**Figure 1 nutrients-13-00416-f001:**
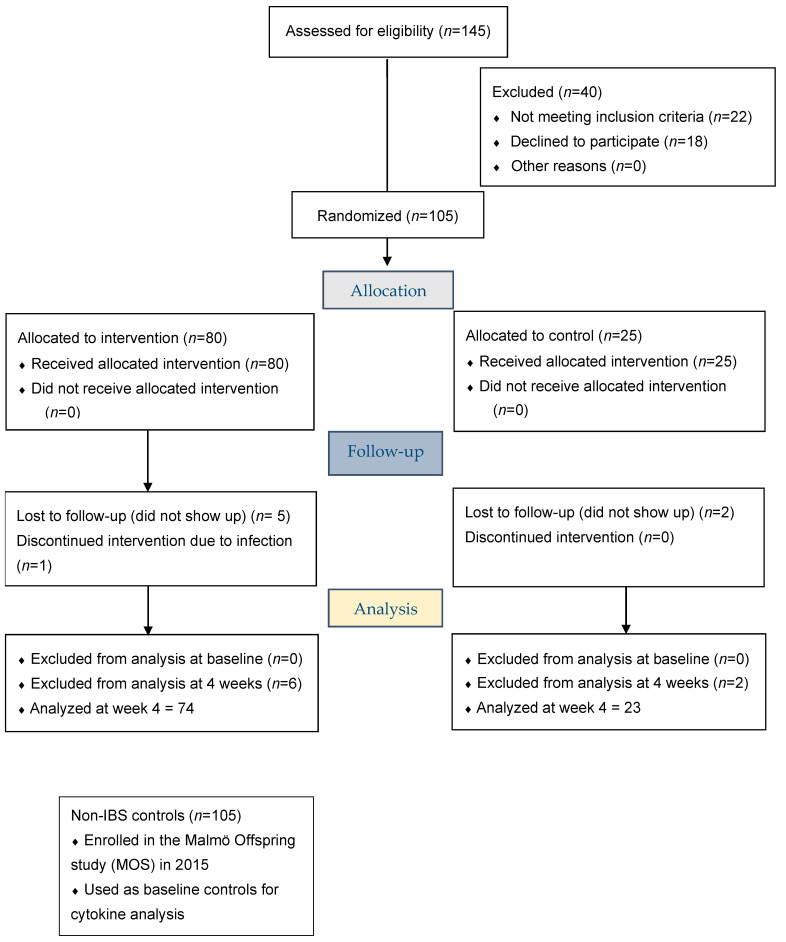
CONSORT 2010 Flow Diagram.

**Figure 2 nutrients-13-00416-f002:**
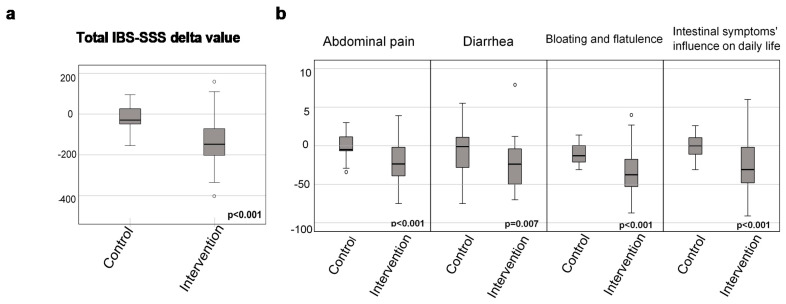
Boxplots representing change in (**a**) total irritable bowel syndrome-symptom severity score (IBS-SSS) [23] and (**b**) visual analog scale for irritable bowel syndrome (VAS-IBS) [24] for the intervention group and controls after the starch- and sucrose-reduced dietary intervention study. Mann–Whitney U test. *p* < 0.05 was considered statistically significant.

**Figure 3 nutrients-13-00416-f003:**
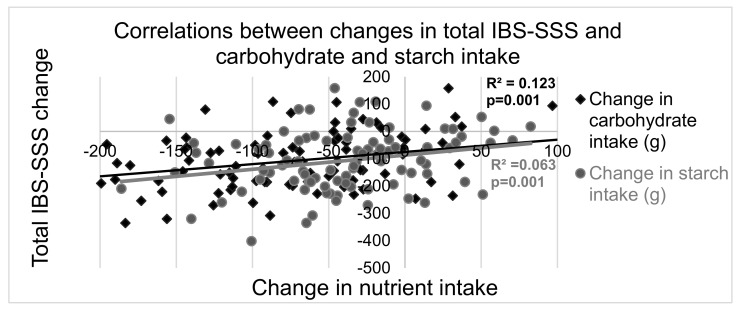
Correlations between changes in total irritable bowel syndrome-symptom severity score (IBS-SSS) [23] and carbohydrate and starch intake (in grams). Calculations were performed with the AIVO Diet computer program [20]. R^2^ = correlation coefficient. *p* < 0.05 was considered significant.

**Table 1 nutrients-13-00416-t001:** Gastrointestinal symptoms and weight before and after the 4-week SSRD intervention.

	Intervention *n* = 80 * (60 Women)	Control *n* = 25 ** (22 Women)		Intervention	Control	
	Median (IQR)	Median (IQR)	*p*	Median of Differences (IQR)	Median of Differences (IQR)	*p*
Age (years)	48 (37–57)	35 (29–50)	0.028			
Disease duration (years)	19 (9.5–29)	12 (10–30)	0.53			
Weight (kg)				−2.1 (−3.0–(−1.0)	0 (−0.5–1.1)	<0.001
Baseline	72 (64–85)	68 (57–75)	0.035			
4 weeks	71 (64–82)	68 (61–77)	0.301			
IBS-SSS total score				−148 (−203–(−72))	−30 (−54–33)	<0.001
Baseline	306 (250–356)	310 (247–351)	0.82			
4 weeks	156 (88–250)	300 (233–331)	<0.001			
Abdominal pain Ref value: 5 (1–13)				−24 (−40–(−2.0))	−5.0 (−7.0–12)	<0.001
Baseline	52 (37–65)	49 (27–63)	0.44			
4 weeks	24 (5.8–43)	50 (32–63)	<0.001			
Diarrhea Ref value: 3 (0–10)				−24 (−49–(−1.8))	−1.0 (−34–12)	0.007
Baseline	57 (18–76)	47 (5.0–71)	0.32			
4 weeks	14 (1.0–33)	24 (1.0–49)	0.27			
Constipation Ref value: 6 (2–16)				−13 (−43–0)	−12 (−40–6.0)	0.41
Baseline	47 (1.0–73)	54 (30–69)	0.76			
4 weeks	18 (0.0–36)	28 (1.0–68)	0.15			
Bloating and flatulence Ref value: 10 (2–23)				−39 (−53–(−18))	−13.0 (−22–1.0)	<0.001
Baseline	77 (59–85)	78 (68–89)	0.38			
4 weeks	28 (11–54)	69 (56–80)	<0.001			
Vomiting and nausea Ref value: 2 (0–4)				−3.5 (−14–1.0)	−4.0 (−16–0)	0.95
Baseline	12 (1.0–37)	29 (5.5–51)	0.13			
4 weeks	3.0 (0.0–24)	12 (2.0–56)	0.04			
Psychological well-being Ref value: 5 (2–15)				−8.0 (−23–2.8)	−1.0 (−14–8)	0.08
Baseline	50 (24–69)	47 (24–71)	0.95			
4 weeks	36 (13–53)	48 (32–60)	0.09			
Intestinal symptoms’ influence on daily life Ref value: 2 (0–14)				−31 (−48–(−2.0))	0 (−12–12)	<0.001
Baseline	72 (52–86)	68 (53–79)	0.54			
4 weeks	35 (20–63)	65 (51–82)	<0.001			
BSFS Average daily stool frequency				−0.1 (−0.5–0.3)	−0.01 (−0.4–0.3)	0.56
10 day-run-in	1.9 (1.2–2.7)	2.0 (1.6–2.4)	0.97			
Intervention	1.6 (1.1–2.6)	1.9 (1.2–2.5)	0.66			

SSRD = starch- and sucrose-reduced diet, IBS-SSS = irritable bowel syndrome-symptom severity score [23], VAS-IBS = visual analog scale for irritable bowel syndrome [24]. * = two missing values (mv) at baseline (three mv for ‘bloating and flatulence’ and ‘stool frequency’) and six mv at week 4. ** = three mv at 4 weeks. Reference values of GI symptoms are from 90 healthy controls [25]. The score ranges between 0–500 for total IBS-SSS and 0–100 for individual GI symptom scores. Stool frequency was calculated from the Bristol Stool Form Scale (BSFS) [26]. Values are presented as median and interquartile ranges (IQR). Mann–Whitney U test. *p* < 0.05 was considered statistically significant.

**Table 2 nutrients-13-00416-t002:** Major food group and fiber intake before and after the 4-week SSRD intervention.

	Intervention *n* = 80 *	Control *n* = 25 **		Intervention	Control	
	Median (IQR)	Median (IQR)	*p*	Median of Differences (IQR)	Median of Differences (IQR)	*p*
Energy (kcal)				−137 (−662–212)	−105 (−389–222)	0.39
Baseline	1660 (1393–2107)	1387 (1202–2027)	0.19			
4 weeks	1472 (1139–1969)	1640 (1216–2169)	0.40			
Carbohydrates (g)				−91 (−142–(−40))	−16 (−73–27)	0.001
Baseline	185 (144–223)	177 (112–208)	0.40			
4 weeks	88 (66–128)	182 (89–224)	<0.001			
Carbohydrates (E%)				−17 (−30–(−5.5))	−6.7 (−18–6.8)	0.007
Baseline	43 (38–49)	43 (37–52)	0.76			
4 weeks	25 (18–36)	42 (32–49)	<0.001			
Protein (g)				8.6 (−16–33)	−0.1 (−11–20)	0.64
Baseline	72 (55–83)	59 (46–71)	0.046			
4 weeks	82 (58–99)	65 (53–82)	0.13			
Protein (E%)				4.7 (0.4–9.8)	1.2 (−1.3–4.7)	0.068
Baseline	16 (14–19)	16 (12–19)	0.35			
4 weeks	21 (19–26)	17 (13–20)	0.006			
Fat (g)				13 (−21–47)	0.3 (−14–39)	0.55
Baseline	65 (45–94)	61 (46–72)	0.39			
4 weeks	72 (56–104)	69 (46–97)	0.26			
Fat (E%)				13 (2.1–21)	7.6 (−7.7–13)	0.018
Baseline	36 (29–43)	34 (27–43)	0.83			
4 weeks	47 (39–55)	38 (31–45)	0.002			
Fiber (g)				−0.6 (−11–4.9)	0.7 (−4.4–3.1)	0.56
Baseline	18 (13–26)	16 (12–22)	0.32			
4 weeks	18 (12–23)	15 (11–22)	0.42			
Fiber (E%)				0.2 (−0.7–0.7)	0 (−0.7–0.3)	0.45
Baseline	2.1 (1.6–2.6)	1.9 (1.4–2.8)	0.89			
4 weeks	2.2 (1.5–2.9)	1.8 (1.4–2.2)	0.052			

SSRD = starch- and sucrose-reduced diet. *n* = Number, E% = energy percentage, g = grams. * = Two missing values (mv) at baseline and six mv at week 4. ** = Three mv at 4 weeks. Nutrient levels were calculated from a single day (day 2) of 4-day food diary registrations; before and at the end of the 4-week dietary intervention. Calculations were performed with the AIVO Diet computer program [20]. Values are presented as median and interquartile ranges (IQR). Mann–Whitney U test. *p* < 0.05 was considered statistically significant.

**Table 3 nutrients-13-00416-t003:** Sugar and starch intake before and after the SSRD intervention.

	Intervention *n* = 80 *	Control *n* = 25 **		Intervention	Control	
	Median (IQR)	Median (IQR)	*p*	Median of Differences (IQR)	Median of Differences (IQR)	*p*
Sucrose (g)				−14 (−32–(−3.5))	−3.8 (−22–4.4)	0.029
Baseline	23 (13–38)	21 (13–43)	0.82			
4 weeks	5.4 (2.4–13)	20 (4.5–36)	0.001			
Sucrose (E%)				−3.8 (−5.8–(−0.8))	−0.9 (−3.8–2.8)	0.008
Baseline	5.5 (4.0–8.8)	6.2 (3.8–11)	0.65			
4 weeks	1.4 (0.8–3.1)	6.3 (2.0–9.0)	<0.001			
Monosaccharides (g)				−5.6 (−18–12)	−6.8 (−18–9)	0.77
Baseline	21 (13–31)	22 (12–31)	0.88			
4 weeks	21 (9.7–29)	17 (6.7–31)	0.80			
Monosaccharides (E%)				−0.8 (−3.2–2.7)	−1.5 (−4.5–2.7)	0.32
Baseline	5.0 (3.3–7.5)	5.1 (3.9–8.2)	0.58			
4 weeks	4.7 (2.5–8.2)	4.4 (2.2–7.6)	0.46			
Disaccharides (g)				−20 (−35–(−4.6))	−10 (−18–7.0)	0.02
Baseline	33 (22–48)	32 (20–51)	0.61			
4 weeks	16 (7.8–27)	28 (18–55)	0.002			
Disaccharides (E%)				−4.4 (−6.1–(−0.9))	−1.8 (−4.8–3.4)	0.042
Baseline	8.1 (6.3–11)	7.3 (5.9–13)	0.86			
4 weeks	4.2 (2.4–7.0)	9.1 (3.9–12)	0.001			
Total sugar (g)				−26 (−54–(−1.0))	−15 (−35–17)	0.13
Baseline	63 (41–89)	58 (42–74)	0.63			
4 weeks	41 (28–58)	58 (21–86)	0.10			
Total sugar (E%)				−4.9 (−9.3–1.4)	−3.0 (−6.5–4.5)	0.34
Baseline	14 (11–19)	14 (11–21)	0.84			
4 weeks	11 (7.5–16)	16 (9.4–20)	0.053			
Starch (g)				−50 (−78–(−23))	−8.3 (−34–34)	<0.001
Baseline	77 (49–116)	71 (43–91)	0.44			
4 weeks	22 (2.6–48)	82 (37–101)	<0.001			
Starch (E%)				−12 (−18–(−0.6))	−1.6 (−9.4–13)	0.001
Baseline	19 (12–25)	17 (14–23)	0.83			
4 weeks	6.1 (0.7–15)	18 (13–26)	<0.001			

SSRD = starch- and sucrose-reduced diet, *n* = number, E% = energy percentage. Total sugar = mono- and disaccharides. * = Two missing values (mv) at baseline and six mv at week 4. ** = Three mv at 4 weeks. Nutrient levels were averaged from a single day (day 2) of the 4-day food diary registrations before/after the intervention. Calculations were performed with the AIVO Diet computer program [20]. Median and interquartile ranges (IQR) are presented. Mann–Whitney U test. *p* < 0.05 was considered statistically significant.

**Table 4 nutrients-13-00416-t004:** Plasma CRP and cytokine concentrations before and after the SSRD intervention.

	Intervention *n* = 80			Controls *n* = 25		
	Median	IQR	*p*	Median	IQR	*p*
CRP Ref: <3.0 mg/L						
Baseline	0.7	0.6–2.2		0.7	0.6–1.0	
4 weeks	0.8	0.6–2.0	0.07	0.8	0.6–1.0	0.62
TNF-α Ref: 0.82 (0.63–1.05)						
Baseline	2.5	2.1–3.0		2.1	1.9–2.8	
4 weeks	2.5	2.1–3.3	0.40	2.5	2.2–2.9	0.003
IFN-γ Ref: 5.54 (3.79–8.83)						
Baseline	2.7	2.1–4.6		2.4	1.5–3.8	
4 weeks	2.8	1.9–4.2	0.89	3.5	2.1–4.5	0.21
IL-6 Ref: 0.70 (0.49–1.06)						
Baseline	0.6	0.4–0.9		0.5	0.4–0.8	
4 weeks	0.6	0.4–0.9	0.68	0.5	0.4–0.8	0.68
IL-8 Ref: 9.79 (7.99–13.21)						
Baseline	12	9.2–14		11	8.1–14	
4 weeks	12	9.4–16	0.017	10	8.4–13	0.29
IL-10 Ref: 0.23 (0.17–0.29)						
Baseline	0.2	0.2–0.3		0.2	0.1–0.4	
4 weeks	0.3	0.2–0.3	0.42	0.2	0.1–0.3	0.07
IL-18 Ref: 946 (585–1640) *	n_0_ = 75; n_4_ = 73			*n* = 22		
Baseline	986	781–1308		978	805–1189	
4 weeks	962	741–1313	0.34	1024	816–1101	0.57

CRP = C-reactive protein, SSRD = starch- and sucrose-reduced diet. Seven missing values (mv) in the intervention group and three mv in the control group at 4 weeks for all cytokines except IL-18. Reference values from laboratory reference values are given for CRP [29] and from 105 non-IBS controls (IL-18 * *n* = 96) for cytokines in pg/mL. Median values and interquartile ranges (IQR) are given. Wilcoxon test. *p* < 0.05 was considered statistically significant.

**Table 5 nutrients-13-00416-t005:** GI symptom score changes (Δ) and Post-hoc analyses of subgroups in the intervention group after the SSRD intervention.

	Δ Mean ± Standard Deviation	*n*	ANOVA with Bonferroni for IBS Subgroup/Non-IBS FGID (*p*)		
			IBS-D	IBS-M	Non-IBS FGID
Total IBS-SSS					
IBS-C	−87 ± 110	13	1	0.054	1
IBS-D	−123 ± 103	21		0.39	1
IBS-M	−177 ± 101	27			1
non-IBS FGID	−139 ± 53	10			
Diarrhea					
IBS-C	24 ± 6.6	13	0.001	<0.001	0.28
IBS-D	−31 ± 25	21		1	0.88
IBS-M	−39 ± 21	27			0.09
non-IBS FGID	−18 ± 32	11			
Constipation					
IBS-C	−30 ± 27	13	<0.001	1	0.75
IBS-D	5.6 ± 16	21		<0.001	0.09
IBS-M	−39 ± 24	27			0.033
non-IBS FGID	−16 ± 24	11			
Bloating and flatulence					
IBS-C	−30 ± 29	13	1	0.12	1
IBS-D	−29 ± 28	21		0.022	1
IBS-M	−51 ± 22	27			0.009
non-IBS FGID	−20 ± 24	10			
Intestinal symptoms’ influence on daily life					
IBS-C	−8.2 ± 33	13	0.49	0.032	1
IBS-D	−27 ± 29	21		1	1
IBS-M	−37 ± 26	27			0.67
non-IBS FGID	−19 ± 38	10			

SSRD = starch- and sucrose reduced diet, IBS = irritable bowel syndrome, IBS-D = diarrhea-predominant IBS, IBS-M = mixed IBS, IBS-C = constipation-dominated IBS, FGID = functional gastrointestinal disease. IBS subgroup diagnosis based on Rome IV criteria [22]. ANOVA with post-hoc Bonferroni correction. *p* < 0.05 was considered statistically significant.

## Data Availability

The data presented in this study are available on request from the corresponding author. The data are not publicly available due to data protection regulation law.

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
