# Peer review of "Assessment of a 4-Week Starch- and Sucrose-Reduced Diet and Its Effects on Gastrointestinal Symptoms and Inflammatory Parameters among Patients with Irritable Bowel Syndrome"

_nutrients, 2021, doi:10.3390/nu13020416_

Round 1

Reviewer 1 Report

See file attached

Author Response

Replies to reviewer no 1

  1. Main concern: the authors have published a previous study on the topic (Nutrients.2019 Jul 20;11(7):1662. doi: 10.3390/nu11071662); this particular study has not been included in the reference list, but it seems the results were obtained from the same participants, with a similar design (including duration of intervention: 4 weeks); the novelty of their present study compared with the previous one, already published, should be clearly stated, is it only the analysis of the plasma levels of cytokines?

Reply: The previous publication (citation no 16 in the present manuscript) focused on frequency of intake of food groups after 2 weeks and extraintestinal symptoms (nausea, difficulty finishing a meal, reflux, belching, headache, back pain, leg pain, muscle/joint pain, urinary urgency, and tiredness). Total IBS-SSS score was also reported, from the same dietary intervention study without any evaluation of specific symptoms or stool consistency. There was no evaluation of nutrient intake.

In the present manuscript focused on 4 week-results, we evaluated

1) both total and specific gastrointestinal symptoms (total IBS-SSS, abdominal pain, diarrhea, constipation, bloating and flatulence, vomiting and nausea, intestinal symptoms’ influence on daily life, stool frequency and consistency according to the Bristol stool scale), as well as

2) participants’ nutrient intakes in macro-and micronutrient composition and

3) systemic cytokine levels before and after the intervention. Further, we correlated changes in specific GI symptom scores with changes in nutrient intake and systemic cytokine levels. 

  1. Title: I would suggest changing “and unaltered inflammatory parameters” to “but

unaltered inflammatory parameters”.

Reply: The title has been changed according to a suggestion from the other reviewer, see page 1, lines 2-5.

  1. Abstract: As in the title, I would suggest changing “and” to “but” in line 30.

Reply: The text has been changed accordingly, see page 1 line 31.

  1. I think the sections “Study design” and “Subjects” should be merged to avoid repetition of information; in addition, figure S1 should be brought to the main text for easier understanding; details should be reviewed to avoid confusion (for example, in line 107 it is stated that 98 patients completed the study, but this is not coherent with what is presented in figure S1; also, it is stated in figure S1 that there were several patients lost after recruitment, but these were not removed from the analysis, please clarify and correct as needed).

Reply: The sections have been merged as suggested and repeated information has been removed. See pages 2-3, lines 84-114 for “study design and subjects”. We included all participants in the baseline analysis, therefore we have written that 0 were excluded. But from calculations at 4 weeks, 6 and 2 were excluded from the intervention and control group, respectively. Hence, 74 participants in the intervention group and 23 participants in the control group were finally included for analysis at 4 weeks, and a total of 97 patients completed the study. The figure S1 has been corrected  as well as brought to the main text, and is now referred to as figure 1.The information has also been corrected in the main text, see page 3, line 101.

  1. Was the level of exercise controlled for? With the wide range of BMI found in the study, it seems necessary to take this into account; this could be a limitation of the study

Reply: The level of exercise was not controlled for; however, we have added information on the level of physical activity of participants at baseline on page 7, lines 282-285 and in supplemental table S3. The level of exercise and BMI did not differ significantly between the intervention and control group at baseline (exercise level was not re-evaluated at 4 weeks). For more information regarding the BMI of participants, please see question No 17 below, where we have answered this thoroughly.

  1. It is clear that the study could not be performed in a blinded fashion, i.e., the patients knew what arm of the study they belonged to; could this limitation be attenuated by applying a cross-over design? Did the authors try any kind of blinding? This problem, probably leading to a relevant placebo effect seem to be the biggest limitation of the study, is it possible to measure this placebo effect in their sample population?

Reply: The placebo effect is certainly of relevance to the study, and is therefore addressed in the discussion section of the manuscript, page 17 lines 560-564. In this pilot-study, we did not perform blinding or use a cross-over design. As stated in the discussion, the use of a control group, who had the same number of visits and contact availability with the study staff as the intervention group, partly adjusts for the placebo effect. As the control group also kept food diaries throughout the study, we were able to observe that the majority continued with their habitual diet. Furthermore, all controls were informed at the study start that they would later, after the study period of 4 weeks, be offered to receive the dietary information to test afterwards. Thus, there should be a placebo effect also in the control group, since they were included in a study and knew they were going to have the same care as the one group which started with the diet.

  1. Line 142: I have noticed that the authors have used FGID and IBS as mutually exclusive diagnostic conditions, but IBS is actually one of the FGID classified according to Rome IV questionnaire, I think confusion should be avoided along the text by using the term “IBS”, on the one side, and that of “non-IBS FGID”, on the other.

Reply: The term “FGID” has been replaced by “non-IBS FGID” throughout the text as suggested.

  1. Healthy controls: it seems that this group was only used for baseline cytokine analyses and to obtain reference values for IBS-related parameters; more data should be provided regarding this group, particularly related with BMI and nutrient intake, similarly to what has been done for IBS patients (this will at least show the impact of Nordic diet on the development of IBS); how were these results obtained? at the same time (i.e., same season of the same year) as those from IBS patients?

Reply: The recruitment of IBS patients was performed between January 2018 and February 2019, thus lasting throughout the year. In the same way, the controls were recruited throughout the year, during 2015. Thus, there was just 2 years between the inclusion of the controls and the inclusion of the IBS patients. The nutritional intake and median BMI of the non-IBS controls have been added to the supplementary material; see table S6.

  1. line 107, please change to “evaluation by a physician”

Reply: This has been changed accordingly (see page 3, line 95).

  1. …line 109, I guess those 105 individuals are IBS patients (not non-IBS controls) and that these were compared with 90 healthy volunteers, please check and correct as

Needed

Reply: The text is referring to two different groups who are not compared with each other.

  1. 105 non-IBS controls (from the MOS registry) used for comparison of cytokine concentrations with the IBS group at baseline and
  2. 90 healthy volunteers from a previous study, used to validate and create reference values for the VAS-IBS symptom scale, ref No 24.

The information regarding the first group has been added and clarified in the text under “Study design and subjects”, page 3, lines 111-114. The information regarding the second group has been moved to the section concerning the VAS-IBS questionnaire (page 5, lines 196-199) for clarity.

  1. …line 127, I think it is not necessary to provide Online resources in two sets, 1 and 2, in addition, the tables mentioned here are actually S4 and S5, I would suggest them to be S1 and S2 because these are the first ones to be mentioned along the main text

Reply: The reference to the supplementary material (all tables and figures in a single document) was incorrect and has been corrected. Supplementary tables are now numbered after their appearance in the main text.

  1. it seems to me that the VAS description is not necessary in IBS-SSS questionnaire section (lines 144-150), it belongs to the next section (lines 151-157), where it is actually described again, please check

and correct as needed

Reply:  The visual analog scale is a measurement tool used both in the IBS-SSS and VAS-IBS questionnaires. The text has been clarified to better indicate this (page 5, line 185).

  1. …line 166, please define “CV”

Reply: This information has been added, see page 6, line 208.

  1. lines 180-183, it does not seem adequate to present the hypotheses here, they are better presented at the end of the introduction

Reply: The hypotheses have been removed from the mentioned paragraph and added to the end of the introduction, see page 2, lines 69-73.

  1. study group size, it seems that there is something missing at the end of the paragraph, how that previous research mentioned by the authors have helped them to determine the group size that should be used in the present study?

Reply:  The paragraph has been extended to further explain the determination of the study group size (page 6, lines 231-237).

  1. …line 196, I think “why” should be changed to “therefore”

Reply: This has been changed accordingly, see page 6, line 241.

  1. Section 3.1.: the BMI range is really wide, from undernutrition to obesity, how

has this impacted the results?

Reply:  Upon reviewing our data, we have found that the BMI reports were incorrect. The median BMI for the group was 24.5 (22.4-27.7) kg/m2 with a range of 16.0–33.2 kg/m2.

Additionally, when re-calculating the basal data; age and disease duration were found to be non-normally distributed. Therefore, medians and interquartile ranges are given for all these parameters (and the BMI range has been corrected), see page 7, lines 269-271.

Considering that the range is not as wide as previously reported, and only one participant was underweight, the BMI is less likely to be a confounding factor. Of the IBS patients included, around 44% were overweight which almost mirrors the frequency in the general population in Sweden, where 51% of the population had a BMI>25 in 2018 (see https://www.folkhalsomyndigheten.se/the-public-health-agency-of-sweden/living-conditions-and-lifestyle/obesity/). Therefore, our numbers are not unusual but in fact keeping with the statistics on a population level. Additionally, since the patients were compared with themselves in paired calculations, this should still not be a major concern and should not influence the results of improvements after the diet. Finally, the median BMI was not significantly different between the intervention and control group at baseline, Table S3.

  1. Section 3.1.: the range of disease duration should be provided, how has this impacted the results?

Reply: The range of disease duration has been added (3.0-60 years) to page 7, line 271. Since the patients were compared with themselves in paired calculations, this should not be a major concern and should not influence the results. There was no significant difference in disease duration between groups, Table 1.

  1. Section 3.2.: a table showing distribution of the 105 patients according to diagnosis should be included, at least as supplementary material; the figures provided in this paragraph (lines 226-233) do not seem to sum 105 patients...

Reply: Two patients had not completed the Rome questionnaire, therefore 2 patients were missing. This has now been added on page 7, line 277. An additional table, supplemental table 3, showing the distribution of IBS patients has been added to the supplementary material.

  1. Table 1: please include sex, age, BMI and duration of disease data; if I understood properly, reference values given in the table for IBS symptoms correspond to those obtained from healthy volunteers, but it would be important to add also the maximum values for each parameter (maybe at the foot of the table); please check for minor typos (i.e., baseline weight in the intervention group, it should be 64-85, not 64.85)

Reply: The requested information (sex, age, BMI and duration of disease) has been added either to table 1 (sex, age and duration of disease) or table S3 (BMI). Information about the score ranges has been included in the foot to table 1, and minor typos have been corrected.

  1. Table 3: I think the same format as in the other tables should be kept also for this one, for easier understanding; this table needs a foot, which I think is currently located in the caption for figure 2, please check and correct as needed

Reply: The formatting errors have been corrected.

  1. Line 324: please check the statement for TNF-alpha (according to table 4, it seems it actually increased in the control group after the 4 weeks)

Reply: This information has been corrected, see page 12, line 396.

  1. Table 4: please check IQR for IL-18 in both groups, it seems the ranges are not complete; please check the foot of the table, there are mentions to axin-1 that do not seem to be relevant here; from line 337, it seems the text corresponds to “main text” not to “table foot”, please check and correct as required

Reply: These formatting errors have been corrected.

  1. I think tables 5.a. and 5.b. should be merged for easier understanding: I think values given in table 5.a. can be included in the first column of table 5.b. (or as a new column in it)

Reply: The tables have been merged according to the suggestion.

  1. As mentioned above, the discussion should include some comment on the diet and energy and nutrient intakes of healthy volunteers; also, please avoid confusion regarding FGID patients (IBS is a kind of FGID)

Reply: The nutrition intake in healthy controls has been added to the text, page 11, lines 355-359, and in supplemental table S6. FGID is now called non-IBS FGID throughout the manuscript, to avoid confusion. The diet and energy and nutrient intakes in non-IBS controls is discussed on pages 15-16, lines 486-489 and page 17, line 566-572.

  1. Line 380: I guess it should read “by way of glucose transporters”, please check

and correct as needed

Reply: The sentence has been clarified (page 15, line 462).

  1. Line 433: please change “graver” to “more severe”

Reply:  This has been changed accordingly (page 16, line 530).

  1. Line 455: the authors claim that “the high compliance observed suggests that the diet is easy-to-follow”, but compliance has not been quantitatively measured (or not stated as such in Results); in order to understand the impact of this, compliance should be compared with that of other diets, in terms of % recruited volunteers that actually followed each of them in the studies

Reply: We have analyzed metabolomics in this cohort, and found that there was a similar composition of metabolomics for groups at baseline, but a great difference, reflecting lower carbohydrate metabolism in the intervention group, after 4 weeks. These data analyses are not shown in the present study because they deserve their own paper.

In the present manuscript, we have added a description of the percentage of patients who lowered their intake of starch and sucrose, respectively, considering both any reduction at all and also >50% reduction from baseline. The differences between groups are clear, especially when considering the frequency of >50% reductions. Please see page 10, lines 346-353.

References:

  1. Please check for completeness, some references lack some details, like the

journal name

Reply: This has been corrected. See page 18-19, Ref No 9 and 17.

  1. Please note reference 8 (“Available from…”) is actually part of reference 7

Reply: This has been corrected, see Ref No 19.

Reviewer 2 Report

The manuscript entitled “High compliance to a 4-week starch- and sucrose-reduced diet results in improved gastrointestinal symptoms and unaltered inflammatory parameters among patients with irritable bowel syndrome” presents interesting issue, but some areas must be corrected.

Major:

  1. The major problem of the presented study is associated with the fact that Authors analysed 2 groups of diverse number of patients, while the control group was very small (disproportionately) – as for the control group n = 25 (for the studied group n = 80). So, the question arises why… and if the sample size was properly estimated (Authors declared that they did not calculate the sample size). Especially if such situation causes that relatively small sample size must be indicated as a significant source of bias. Taking this into account, it may be hard to conclude, as the results that are observed are biased.
  2. Authors declare that their patients had their “food diary registrations”, but based on the presented information, it is hard to guess if it was dietary recall or dietary record, as no detailed methodology is described.

General:

The manuscript is shabbily prepared. It should be formatted according to the instructions for authors.

Authors should apply unambiguous numbering of references, as for the time being their references have various numbers, so it is hard to guess which one is in fact referred in the text.

Authors in their text refer supplementary materials, but I suppose that they refer wrong tables – e.g. Authors indicated that patients were provided information about recommended products and they refer table S3 (Correlations between changes in gastrointestinal symptoms and nutrient levels).

Title:

Authors should formulate their title to present what was studied (e.g. “analysis of…”, “assessment of…”), not the conclusion from their study. Especially if their current title is more catchy than justified.

Abstract:

Lines 15-16 – this general statement is not justified

Authors should briefly describe applied intervention.

Authors should present specific numeric values that were obtained and the results of the conducted statistical analysis (p-Values).

Authors should formulate brief applicative conclusion (e.g. what should be recommended?).

Introduction:

Authors present a number of basic and even very trivial information that should not be presented in a scientific manuscript (e.g. “Starch, found primarily in grains, root vegetables and beans, and sucrose, found in confectionary and sweetened foods and beverages, are common carbohydrates in the human diet [7]”) – Authors should be aware that they do not prepare the basic manual for students, or column of the newspaper, but a scientific paper that should be interesting for researchers from the area of food and nutritional sciences, so they should understand that their readers will have the nutritional knowledge.

Authors should not focus on their own previous research, but they should rather present more universal perspective based on the studies of various research teams.

Lines 71-79 – Authors should briefly present the aim of their study and all the redundant information they may transfer to Materials and Methods Section.

Materials and Methods:

Authors should clearly specify the inclusion and exclusion criteria.

Authors should describe the program with all the necessary details. Authors should be aware that this issue is the most important for the readers of the Nutrients journal.

Authors should describe in details the applied methodology of dietary assessment – authors should describe all the details (e.g. how were the sample sizes estimated, how were products described, etc.)

“Tests for normality were performed by the Kolmogorov-Smirnov test and examining data graphically.” – why did Authors apply 2 various methods and which was the basis for their conclusions?

Results:

Figure 1 – results should be rather presented as table to be easier to follow

Authors should not reproduce in the text information that are already presented in tables.

Discussion:

Authors should not discuss the issue of gluten, as it was not studied in their research.

Authors should in their discussion include 3 areas: (1) compare gathered data with the results by other authors, (2) formulate implications of the results of their study and studies by other authors, (3) formulate the future areas which should be studied.

Conclusions:

Authors should briefly formulate conclusion from their study.

Author Contributions:

It seems that contribution of some Authors (EL, BR) was only minor and they did not participate in preparing manuscript. There is a serious risk of a guest authorship procedure which is forbidden. In such case they should be rather presented in Acknowledgements Section and not be indicated as authors of the study.

References:

Authors should include adequate references, while self-citations should be avoided, as they are not adequate (Authors included 6 self-citations – 14% of all the references).

Author Response

Replies to reviewer no 2

Major:

  1. The major problem of the presented study is associated with the fact that Authors analysed 2 groups of diverse number of patients, while the control group was very small (disproportionately) – as for the control group n = 25 (for the studied group n = 80). So, the question arises why… and if the sample size was properly estimated (Authors declared that they did not calculate the sample size). Especially if such situation causes that relatively small sample size must be indicated as a significant source of bias. Taking this into account, it may be hard to conclude, as the results that are observed are biased.

Reply: Although not discussed in the present paper, the complete study scope included genetic testing for a rare SI functional gene variant, which influenced the intervention/control group ratio as we also planned to stratify the intervention group according to gene variant. Therefore, the two groups were disproportionate in size. However, the participants were compared with themselves, and the differences between baseline and 4 weeks were marked, which supports our conclusions.

  1. Authors declare that their patients had their “food diary registrations”, but based on the presented information, it is hard to guess if it was dietary recall or dietary record, as no detailed methodology is described.

 Reply: This information has been added (page 5, lines 161-169).

General:

The manuscript is shabbily prepared. It should be formatted according to the instructions for authors.

Authors should apply unambiguous numbering of references, as for the time being their references have various numbers, so it is hard to guess which one is in fact referred in the text.

Reply: The formatting errors have been corrected.

Authors in their text refer supplementary materials, but I suppose that they refer wrong tables – e.g. Authors indicated that patients were provided information about recommended products and they refer table S3 (Correlations between changes in gastrointestinal symptoms and nutrient levels).

Reply: This has been corrected accordingly. Additionally, the numbering of supplementary tables has been changed according to their appearance in the main text.

Title:

Authors should formulate their title to present what was studied (e.g. “analysis of…”, “assessment of…”), not the conclusion from their study. Especially if their current title is more catchy than justified.

Reply: The title has been changed according to the suggestion, see page 1, lines 2-5.

Abstract:

  1. Lines 15-16 – this general statement is not justified
  2. Authors should briefly describe applied intervention.
  3. Authors should present specific numeric values that were obtained and the results of the conducted statistical analysis (p-Values).
  4. Authors should formulate brief applicative conclusion (e.g. what should be recommended?).

Reply: (1) The general statement has been removed, (2) the intervention has been described and (3) specific numeric values have been added. See page 1, lines 14-31 (due to the word limit constraints, we have focused on the most important parts of the feedback here).

Introduction:

Authors present a number of basic and even very trivial information that should not be presented in a scientific manuscript (e.g. “Starch, found primarily in grains, root vegetables and beans, and sucrose, found in confectionary and sweetened foods and beverages, are common carbohydrates in the human diet [7]”) – Authors should be aware that they do not prepare the basic manual for students, or column of the newspaper, but a scientific paper that should be interesting for researchers from the area of food and nutritional sciences, so they should understand that their readers will have the nutritional knowledge.

Reply: The paragraph has been removed from the introduction as suggested (page 2, lines 50-51) and from the text describing the structure of starch in the discussion section as well (page 15, line 444).

Authors should not focus on their own previous research, but they should rather present more universal perspective based on the studies of various research teams.

Reply: References are made to previous research by others throughout the introduction. The only reference to previous work specifically by our research group is used to describe the results from the same study at 2 weeks, which we feel is relevant information for the reader. There is no previous study examining the SSRD in IBS patients, and previous research even on carbohydrate-restricted diets in IBS is indeed scarce as stated. If the reviewer has any specific example of studies relevant to incorporate into the introduction, we’d gladly consider doing so.

Lines 71-79 – Authors should briefly present the aim of their study and all the redundant information they may transfer to Materials and Methods Section.

Reply: The paragraph describing study aims has been shortened (page 2, lines 73-76) and redundant information about the correlations can be found in the methods section (see statistical analyses-non-parametric statistical analyses).

Materials and Methods:

Authors should clearly specify the inclusion and exclusion criteria.

Reply: A description of the inclusion and exclusion criteria has been added to the main text, see page 3, lines 96-101.

Authors should describe the program with all the necessary details. Authors should be aware that this issue is the most important for the readers of the Nutrients journal.

Reply: The study design is described on page 2-3, lines 84-114. The individual questionnaires included are then further explained in detail, page 5-6, lines 179-203. Information regarding the food dietary registrations and inclusion/exclusion criteria has been expanded upon according to the suggestions made.

Authors should describe in details the applied methodology of dietary assessment – authors should describe all the details (e.g. how were the sample sizes estimated, how were products described, etc.)

Reply: The patients recorded, in a free-writing structure on an empty paper, both time point and volume/ amount of intake for all food items consumed. They described in detail all food items ingested, including  fat% for dairy products, fiber content (%) for bread, cacao (%) for chocolate and information on the type of soda (sugar-free or regular) consumed.

This information is now added on page 5, lines 161-169.

“Tests for normality were performed by the Kolmogorov-Smirnov test and examining data graphically.” – why did Authors apply 2 various methods and which was the basis for their conclusions?

Reply: The K-S test was used primarily, and in some cases, data was also examined graphically for confirmation. This has been changed to simplify understanding for the reader (page 6 line 239).

Results:

Figure 1 – results should be rather presented as table to be easier to follow

Reply: The  results in figure 1 are presented in table 1 as well (see medians of differences).

Authors should not reproduce in the text information that are already presented in tables.

Reply: The information has been removed from the text when already presented in tables, see page 8, lines 310-319 and page 12, lines 372-374.

Discussion:

Authors should not discuss the issue of gluten, as it was not studied in their research.

Reply: In the discussion section, we propose a theory describing a possible mechanism behind the positive effects of the SSRD. During the SSRD intervention, participants greatly decreased their carbohydrate and starch intake, and in all likelihood their wheat intake (as wheat-containing products were discouraged). Considering this and the fact that non-celiac wheat sensitivity is a well-known proposed mechanism behind IBS, we find it relevant to discuss in the context of this study. We have, however, shortened and toned down the paragraph, to be more speculative in nature. See page 16, lines 506-509 and 520-521.

Authors should in their discussion include 3 areas: (1) compare gathered data with the results by other authors, (2) formulate implications of the results of their study and studies by other authors, (3) formulate the future areas which should be studied.

Reply: The discussion has been revised and re-structured in its entirety to better fulfill the criteria outlined in the above points. For point No (3), see page 17-18, lines 576-583.

Conclusions:

Authors should briefly formulate conclusion from their study.

Reply: A conclusion has been added, see page 17, lines 574-576.

Author Contributions:

It seems that contribution of some Authors (EL, BR) was only minor and they did not participate in preparing manuscript. There is a serious risk of a guest authorship procedure which is forbidden. In such case they should be rather presented in Acknowledgements Section and not be indicated as authors of the study.

Reply: EL contributed greatly to the study by going through all dietary records and performing the nutrient calculations for all included patients. BR helped include all patients, and was the main contact for participants throughout the 2-year study. She was solely responsible for blood sampling and also performed laboratory analyses together with CN. We therefore feel it is justified to keep them as authors of the present study. Additionally, both authors (EL and BR) have read the manuscript and given feedback, and this information has now been added to the author contribution paragraph (page 18, lines 594-595).

 References:

Authors should include adequate references, while self-citations should be avoided, as they are not adequate (Authors included 6 self-citations – 14% of all the references).

Reply: Self-citations have been reviewed, and removed from the main text when possible. Most of the self-references, however, were cited because they describe the validated questionnaires, and the control material used in the present study. Furthermore, the previously published manuscripts describing the results from the present trial should be mentioned, as they put our results into context. 

Round 2

Author Response

Dear Editor and reviewer,

Thank you for giving us a second opportunity to revise the manuscript, nutrients-1054675, entitled “Assessment of a 4-week starch- and sucrose-reduced diet and its effects on gastrointestinal symptoms and inflammatory parameters among patients with irritable bowel syndrome”. We have now tried to further explain and describe the program and method used. All changes are marked in yellow.

Reviewer no 1

The authors have adequately addressed many of my concerns, but there are still a few issues

that deserve attention.

Major:

- Why were not the same healthy controls used for all the studies? You have 105 from the MOS, from whom baseline cytokine levels and nutrient intakes were recorded, and then 90 healthy volunteers from staff at the hospital to obtain VAS-IBS reference data… Did I understand well? I think this should be clearly explained and stated as a possible limitation of the study. These 90 volunteers should be mentioned in the general section about study design and subjects. Both types of healthy controls (105 and 90) should be mentioned in Figure 1, including the parameters measured in each case, together with recruitment details (center, years).

Reply: The VAS-IBS is an established, validated questionnaire; however, it is not as well-known as the IBS-SSS which is more widespread and commonly used. Therefore, we thought it was important to give some background information and explain how the reference values were obtained. We feel, however, that it would be confusing for the reader to add these controls to figure 1 since they were only used for obtaining reference values, in a previous study. In the study referenced as No 25,  healthy individuals were invited to complete the VAS-IBS. Although they were healthy and did not have any abdominal complaints, they did not rate 0 in all vas scales, as can be seen in table 1.

The MOS was a study including several thousands of subjects from the general population with a large questionnaire with many questions, including the VAS-IBS. We first performed a pilot study of 500 subjects before enrolling further subjects. They were confused when they had to answer questions about GI complaints when they did not have any GI symptoms. Thus, to make it easier in that study, we added a question before the VAS-IBS, where participants were asked to answer a yes/no question regarding presence of GI complaints during the past 2 weeks. Those who answered no, were recommended to not complete the VAS-IBS and instead proceed to the next part of the questionnaire. Thus, the 105 participants from the MOS used in the present study technically all have   a VAS-IBS score of zero. We therefore thought that it was more correct to use the other 90 healthy controls who had actually filled out the visual analog scales. We have now added this information in the method section, page 3 line 126-129, page 6, lines 201-205 and page 7, 246-248. We have also added information about the 105 non-IBS controls to figure 1.

- Recruitment: Please clarify and be coherent in the text: i.e., see page 3, line 88 (according to this, recruitment was from one center between 2015 and 2017, and another between 2016 and 2017), which does not correspond to the answer given by the authors... The information regarding recruitment of these non-IBS controls should be added to the text (years in which they were recruited for the MOS registry). From further information provided in the text, it seems that these controls were not only used to obtain baseline cytokine analysis. If they underwent questionnaires and other kind of evaluation, this should be mentioned at the end of the paragraph (page 3, lines

111-114). Please check and correct as required.

Reply:

  1. The first round of recruitment took place in 2015-2017 and 2016 and 2017, respectively. This means that the patients were diagnosed with IBS at their respective healthcare center at some point during this time.

We received lists of patients centrally from the county council and then contacted them by mail and phone.

  1. The second round of recruitment: January 2018-February 2019 was the time period when we included them in the current study.

We have now clarified this information on page 3, lines 104-105 .

More information is given about MOS and the non-IBS controls, page 3, last section. However, these controls were only used for comparison of cytokine levels. We also used sex and BMI to adjust for differences between groups. Additionally, we have added information about nutrient intake in these controls according to the previous suggestions from major revision round 1 (table S7).

- Figure 1: Just to make it clearer: please add the following in the last box of the two groups: Analyzed at week 4 = 74 (left); Analyzed at week 4 = 23 (right=). The total number of participants that ended the study is 98, please check the figure given in page 3, line 101 (currently, 97 is the figure provided). In addition, please check the location of the box showing the allocation for control group (figure 1), it should be moved a little bit downwards.

Reply:  The requested information has been added to, and formatting errors have been corrected in, figure 1. The total number of participants who completed the study was 97, so this information is correct.

- Was the CRP and cytokine analyses method the same for IBS patients and healthy controls? Or do the methods described apply only for IBS patients (with/without intervention?)

Reply: CRP was only analyzed in IBS patients, both intervention group and control group. Cytokine levels were analyzed at the same time in the laboratory with the same lot numbers and by the same methods. This is now further explained on page 7, lines 259-260.

- Line 356: in addition to the dietary intake, also BMI that was different between IBS and non-IBS patients at baseline, this should be mentioned here and taken into account probably in the discussion too. It is interesting to see that IBS patients showed lower energy intake but higher BMI, how can this be explained? Would it be due to differences in physical exercise at baseline between these two groups?

Reply: There may be differences in the food registry. In our current study of IBS, all wrote on a paper what they ingested, in a free writing structure with given amounts/volumes. In MOS, they had a web-based registry. To assess dietary intakes and habits, a web-based 4-day food record designed by the Swedish National Food Agency (called “Riksmaten 2010” in Swedish) was used to capture absolute dietary intake during a four-day observation period. Before the food recording, the MOS participants were tutored via an instruction video (https://www.youtube.com/watch ?v=DB3bzD0FJMg) to register everything they ate and drank during four consecutive days, starting the day after their first visit to the clinic, in order to get a representation of all weekdays in the cohort. To help the participants to register as correctly as possible, they were provided with a notebook and a photo book with portion sizes, identical to what was found on the online registration page.

The differing ways of recording dietary intakes may lead to small differences in registration between the IBS patients and MOS controls. It is also inherently difficult to estimate energy intake, and those with a higher BMI have a higher tendency to misreport their food intake. It may be that patients with IBS who are going to be included in a dietary trial underreport their intake of prohibited foods before entering the study.

We have looked at the data describing physical activity, and although the data are not comparable (statistically) between the two groups due to differing categorical variable measurements being used, it appears the MOS controls had higher levels of physical activity. Since more healthy individuals are part of the MOS control cohort, they might be expected to have better habits, both when it comes to eating patterns as well as physical activity. Additionally, healthy individuals who volunteer to enter a study such as the MOS, might be expected to be more health-aware and health-concerned than the average person.

To conclude, differences between the groups in dietary registration method as well as physical activity levels might very well help to explain our results. Controls were age-matched, and we also adjusted for differences in BMI and sex in the cytokine analyses. Although consideration of other factors is indeed interesting and of relevance, our primary focus by using the MOS controls was to compare cytokine levels at baseline, not to use MOS controls for a general cross-sectional comparison with the IBS patients. We have added information about physical activity and dietary registrations, in the context of possible limitations of the cytokine analyses, in the discussion section, page 19, lines 656-662.

- Section 3.1.: I still think there are some problems here: The range is still too wide and there are some incoherencies, since according to your previous answer it is from 16.0 to 33.2, but to 39.8 in the text. Also, I think the basal characteristics of all participants should be mentioned here, not only the characteristics of the 80 patients that underwent the intervention. A mention to table S3 should be included in this section too. In addition, there are some incoherencies between the data provided here and that provided in table S3, please check and correct. The paragraph included at the end of section 3.2 (lines 281-285) should be moved to section 3.1. Please merge and present the data in a less confusing way.

Reply: The correct range is 16.0 – 39.8 as indicated in the text, and these numbers are also the basis for the reasoning in our previous answer (the number of 33.2 had not been changed accordingly). The range is wide, but IBS is a chronic disease that lasts for years. There was no difference in disease duration between the two groups as shown in table 1. The first paragraph under section 3.1. describes the whole cohort (n=105), which is now more clearly indicated (page 9, line 323). The differences between baseline characteristics of the control and intervention group are now described on page 9, lines 326-330, and this paragraph has been moved according to the suggestion. The data presented in table 1 and table S3 describe differences between the control and intervention group and are referred to in the same paragraph (lines 326-330). Since the data in the first part of section 3.1 (page 9, lines 323-325) concerns the whole group, we do not refer to any tables here.

Thank you for the suggestion to move the paragraph, it is much clearer now.

- Section 3.1. Despite the fact that paired calculations were performed to reduce the impact of individual effects, the wide range of disease duration may have influenced the results. Was there any control for treatment received, either pharmacological or psychological?

Reply: The occurrence of comorbidities and drug treatment of IBS patients did not differ significantly between the intervention and control  group. Information is now added in the text (page 9, lines 331-333) and in supplementary table S4.

- Section 3.2: Table S3 should be mentioned in this first paragraph when describing the distribution of participants according to the different diagnoses (type of FGID, non-FGID). Regarding the two missing questionnaires, please specify that these were from the intervention group (line 277). Please add the % after “Forty-eight subjects” for coherence with other descriptions.

Reply: Table S3 is now referred in section 3.2. The two participants with missing values of ROME IV questionnaire were in the intervention group, and this information has been added to the text in section 3.2. See page 9, lines 339-341.

- Table 1 should be shown before figure 2 (it can be moved to appear on page 8 instead of 9).

Reply: The table and figure have been switched in order of appearance according to the suggestion.

- Table 3: I still think this can be improved and make it in the same format as other previous tables, particularly regarding the p-values. Can this table not be presented in a similar way as Table 2, with at least one less p-value column and less figures on the last column?

Reply: We have now moved the p-values of comparisons between baseline and 4 weeks within the groups (Wilcoxon test) to the text (page 11, lines 406-408), and only show the p-values for differences between groups - at baseline and 4 weeks, as well as in distribution of median differences – in table 3, similarly to the other nutrient-related tables.

Minor:

- Page 3, line 92: please correct to “patients”

Reply: This has been corrected accordingly.

- Page 3, lines 93-94: please check because 82+28=110, not 105

Reply: Out of all IBS patients included (n=105), 82 were females and 23 were males. Also, 77 patients were recruited from primary care and 28 patients were recruited from the tertiary health care center. This has been clarified in the text. See page 3, line 114-116.

- Page, 3, line 98: please define IBS-SSS here (first time of occurrence) and not in line

103

Reply: This has been changed accordingly. See page 3, line 97.

- Page 3, line 101: please move “Ninety-seven (97) patients completed the study” to the

end of next paragraph (after “Blood samples were collected… until analysis”).

Reply: The sentence has been moved to the end of the paragraph. See page 3, line 121.

- Page 5, line 162: please check if it should be changed to “free writing”

Reply: This has been changed accordingly. See page 6, lines 207-208.

- Page 6, line 203, please define the abbreviation BSFS

Reply: The abbreviation is now defined, see page 3 line 98.

- Page 6, line 220: please check to “were” (for levels)

Reply: This has been changed accordingly. See page 7, line 271.

- Page 8, line 291: please check that the reference given here (22) is the correct one (it should be about IBS, but it is about MOS registry); it seems ref.22 is also mistakenly mentioned in line 324; in line 325, it seems that the reference should not be 19 (maybe 26) and in line 327 it seems that the ref. should be 27 (not 26)

Reply: We apologize for these errors. The citations have now been corrected. See page 9, line 350 and page 10, lines 367, 369 and 371.

- Page 8, lines 294-296: Why was diarrhea not mentioned here? It showed a much smaller p-value than vomiting and nausea, for example.

Reply: In the mentioned paragraph, we describe the difference between absolute scores at 4 weeks for the intervention and control group (p=0.27). The median of differences (from baseline to 4 weeks) were significantly different between the groups (p=0.007), as is described on page 11, lines 374-378. This is also illustrated in figure 2b.

- Page 8, line 297: I think you should mention “the remaining VAS-scores” that did not differ significantly between groups. Furthermore, this sentence is repeated at the end of the paragraph, repetitions should be avoided.

Reply: The VAS-IBS scores have been specified and the repetition of phrasing has been removed. See page 9, lines 356-357.

- Please check the p-value given for the difference between intervention and control groups for Vomiting and nausea in Table 1 (it is stated to be 0.95)

Reply: The p-value has been checked and it is accurate.

- Please check p-value also for Fiber in Table 2 (it is mentioned to be 0.007)

Reply: The column with p-values was incorrect in its entirety. We apologize for this. The p-values have now been corrected. The changes of these p-values did not affect the conclusions previously drawn from the results, but rather support them further.

- Table S7: please check if N=96 is correct (would it not be 97?)

Reply: There was one patient who did not fill out the dietary registration at 4 weeks. So, there was one missing value for all correlations and n=96. This has been clarified in the table foot.

- Line 437: it might be useful to start the sentence as "However, the SSRD..."

Reply: This has been changed accordingly. See page 16 line 523.

- Line 463: please define SGLT-1

Reply: The term has been defined. See page 17 line 549.

Reviewer 2 Report

The manuscript entitled “Assessment of a 4-week starch- and sucrose-reduced diet and its effects on gastrointestinal symptoms and inflammatory parameters among patients with irritable bowel syndrome” presents interesting issue, but some areas must be corrected.

Major:

  1. The major problem of the presented study is associated with the fact that Authors analysed 2 groups of diverse number of patients, while the control group was very small (disproportionately) – as for the control group n = 25 (for the studied group n = 80). So, the question arises why… and if the sample size was properly estimated (Authors declared that they did not calculate the sample size). Especially if such situation causes that relatively small sample size must be indicated as a significant source of bias. Taking this into account, it may be hard to conclude, as the results that are observed are biased.
  2. Authors should describe the program with all the necessary details. Authors should be aware that this issue is the most important for the readers of the Nutrients journal.

General:

The manuscript is still shabbily prepared. It should be formatted according to the instructions for authors.

Authors should in their text refer all the supplementary materials.

Abstract:

Authors should briefly describe applied intervention – with more details not only “receiving verbal and written dietary advice”

Authors should formulate brief applicative conclusion (e.g. what should be recommended for patients?).

Materials and Methods:

Authors should describe the program with all the necessary details. Authors should be aware that this issue is the most important for the readers of the Nutrients journal.

Results:

Figure 2 – results should be rather presented as table to be easier to follow

References:

Authors should include adequate references, while self-citations should be avoided, as they are not adequate (Authors still included 6 self-citations – over 13% of all the references).

Author Response

Reviewer no 2

Major:

  1. The major problem of the presented study is associated with the fact that Authors analyzed 2 groups of diverse number of patients, while the control group was very small (disproportionately) – as for the control group n = 25 (for the studied group n = 80). So, the question arises why… and if the sample size was properly estimated (Authors declared that they did not calculate the sample size). Especially if such situation causes that relatively small sample size must be indicated as a significant source of bias. Taking this into account, it may be hard to conclude, as the results that are observed are biased.

Reply: As we answered last revision round, and have written on pages 7-8 (under “study group size”), the intention was to include genetic testing. Since the geneticist wanted as many patients as possible in the intervention group to perform calculations regarding the effect of the diet in relation to functional gene variants, there was a disproportionate balance between the groups.

We have added comparisons between the groups regarding baseline characteristics such as age, weight, sex, BMI, disease duration, physical activity and IBS subgroup (table 1 and supplemental table 3). Age and weight differed, with lower median age and weight in the control group, but without difference in BMI, sex and disease duration. Importantly, both groups in the study serve as their own controls, as outcomes are measured over time. To further compensate for the baseline differences, we compared the differences not only between the groups but also within the groups (Wilcoxon test). We cannot affect the intervention to control ratio now afterwards, but have added it as a limitation in the discussion, page 19, lines 652-656.

Although not mentioned in this manuscript, we have also examined metabolomic data in these patients, which revealed a clear difference between the metabolic profiles of the intervention vs control group at 4 weeks. Thus, we have reason to believe that there is a true difference between groups, due to the applied intervention (the SSRD) primarily, and not to other factors. This is a pilot trial, and the results need to be confirmed in another larger study, and the diet needs to be compared with other established dietary treatments for IBS.

  1. Authors should describe the program with all the necessary details. Authors should be aware that this issue is the most important for the readers of the Nutrients journal.

Reply: We have added more information to the method section concerning the following:

  • The recruitment process (page 3, lines 88-105)
  • Reasons for exclusion (page 3, lines 112-113)
  • The MOS control cohort (page 3, lines 126-129, page 6, lines 199-205 and page 7, lines 246-248)
  • Dietary advice (page 4, lines 137-146 and 154-159)
  • The study questionnaire (page 6, lines 198-205)
  • CRP and cytokine analyses (page 7, lines 256 and 259-260)

If there is anything else the reviewer finds is missing in the methods section, we are happy to revise further and add this into the text.

General:

The manuscript is still shabbily prepared. It should be formatted according to the instructions for authors.

Authors should in their text refer all the supplementary materials.

Reply: All supplementary tables and figures are referred to as follows:

Tables S1-S2: page 4, line 152

Table S3: page 9, line 330

Table S4: page 9, line 333

Table S5 and S6: page 11, lines 392, 395, 397 and 399

Table S7: page 12, line 431

Table S8: page 13, lines 444, 445, 448 and 451

Figure S1: page 3, line 87

We have scrutinized the text thoroughly and corrected all the things we have been able to see. The tables have been formatted according to the instructions for authors.

We have also added subsections to the methods section (2.1, 2.2 etc.) for clarity. We apologize for the mistakes in the former version. You are welcome to address further errors so we can revise accordingly.

Abstract:

Authors should briefly describe applied intervention – with more details not only “receiving verbal and written dietary advice”

Authors should formulate brief applicative conclusion (e.g. what should be recommended for patients?).

Reply: The requested information has been added to page 1, lines 18-19 (dietary advice) and 33-35 (applicative conclusion).

Results:

Figure 2 – results should be rather presented as table to be easier to follow

Reply: All values from Figure 2 are presented in Table 1. We just wanted to make it easier to see the results by depicting them visually in a figure.

References:

Authors should include adequate references, while self-citations should be avoided, as they are not adequate (Authors still included 6 self-citations – over 13% of all the references).

Reply: We have now excluded citation No 17

All 5 references included are adequate as we can see it.

Nilholm et al. 2019: It describes the previous publication which may be more detailed in some sections. This reference also include blood samples which showed lower circulating values of iron and vitamin D in IBS patients, which is interesting since we could see in this present study that they had a low intake of food containing minerals/vitamins.

Ohlsson et al. 2017 describes the 105 non-IBS controls used for measurements of cytokines in plasma

Bengtsson et al. 2007 describes the validation of the VAS-IBS instrument

Bengtsson et al. 2013 describes reference values of VAS-IBS in the 90 healthy control group and also validated that the VAS-IBS could be used to measure changes over time. As we have explained above, we needed two different control groups of healthy volunteers. In the study referred to No 25, we invited healthy individual to complete the VAS-IBS. Although they were healthy and did not have any abdominal complaints, they did not rate 0 in all VAS scales, as can be seen in table 1. In MOS, which was a study including several thousands of subjects from the general population with a large questionnaire with many questions including the VAS-IBS, we first performed a pilot study of 500 subjects before enrolling further subjects. They were confused when they had to answer questions about GI complaints when they did not have any GI symptoms. Thus, to make it easier in that study, we added a first question before VAS-IBS, where we asked whether they had had any GI complaints in the past 2 weeks. They had to answer “yes” or “no” on this question. Those who answered no, were recommended to not complete VAS-IBS but to go to next questions. Thus, all those 105 participants from MOS have 0 in their VAS-IBS scales. We therefore thought that it was more correct to use the other 90 healthy controls who had actually completed the scales. We have now added this information in the method section, page

Nilholm et al. 2018 explains the rationale for the study size in the present cohort. Since it was a first pilot study, we could not perform any power calculation. However, we have performed another dietary intervention previously and could use that study to estimate the sample size needed to assess any differences between groups regarding symptoms and cytokine levels. Nilholm et al. 2018 is the only study that reported the cytokine effect of the previous dietary intervention. This is important to describe under statistical calculations.

Thus, we do not think that we can delete any of these references, since we ought to show all data and subjects included.
